

# Assessing reliability of hydrological simulations through model intercomparison at the local scale in the Everest region.

Judith Eeckman[a], Santosh Nepal[b], Pierre Chevallier[a], Gauthier Camensuli[a], Francois Delclaux[a], Aaron Boone[c], and Anneke De Rouw[d]

[a]Laboratoire HydroSciences (CNRS, IRD, Universite de Montpellier) CC 57 - Universite de Montpellier 163, rue Auguste Broussonnet 34090 Montpellier, France;
[b]International Centre for Integrated Mountain Development (ICIMOD), GPO Box 3226, Kathmandu, Nepal;
[c]CNRM UMR 3589, Meteo-France/CNRS, Toulouse, France;
[d]Institut d'Ecologie et des Sciences d'Environnement de Paris (IRD, UPMC), 4place Jussieu, 75252 Paris cedex 5, France.

*Correspondence to:* Judith Eeckman (judith.eeckman-poivilliers@univ-montp2.fr)

**Abstract.** Understanding hydrological processes of high-altitude areas is vital because downstream communities depend on water resources for their livelihood. This paper compares the hydrological responses at the local scale of two models using different degrees of refinement to represent physical processes in sparsely instrumented mountainous Himalayan catchments. Two small catchments located in mid- and high- mountain environments were chosen to represent the very different climatic and
5   physiographic characteristics of the Central Himalayas in the Everest region of eastern Nepal. This work presents the novelty of applying, at a small spatio-temporal scale and under the same forcing conditions, a fully distributed surface scheme based on mass and energy balance equations (ISBA surface scheme), and a semi-distributed calibrated model (J2000 hydrological model). A new conceptual module coupled to the ISBA surface scheme for flow routing is presented. The results show that both models describe the evapotranspiration, quick runoff and discharge processes in a similar way. The reliability of the simulations
10  for these variables can therefore be considered as satisfactory. The differences in the structure and results of the two models mainly concern the water storages and flows in the soil, in particular for the high-mountain catchment. This conclusion suggests that the uncertainty associated with model structure is significant for water storage and flow in the soil.

    entral Himalayas; ISBA surface scheme; J2000 model ; water budget at the local scale; structural uncertainty;



## Introduction

Modelling hydro-climatic systems for a Himalayan catchments is particularly challenging because of the double-edged situation of highly heterogeneous and sparsely instrumented catchments. On the one hand, sharp topographic variations in this region result in extreme climatic heterogeneities (Barros *et al.*, 2004; Anders *et al.*, 2006) and on the other hand, the high-altitude areas have limited hydro-meterological monitoring devices. A combination of these issues critically limits the representation of hydrological responses at regional scales in the Himalayan region.

The central part of the Hindu Kush Himalaya region ranges from the Terai agricultural plain in the South to the highest peaks in the world to the north (FIGURE 1). The two main driving climatic processes are the summer Indian monsoon, which contributes approximately $80\%$ of the total annual precipitation over the central Himalayan range (Bookhagen and Burbank, 2006; Dhar and Rakhecha, 1981), and winter precipitation arising from westerlies (Lang and Barros, 2004).

Limited access and physical constraints stemming from the region's steep topography explain that the density of meteorological stations is particularly low in the Himalayan region. Recorded time series are more often short in duration and associated with significant uncertainties (Salerno *et al.*, 2015). Moreover, most of the stations are located in river valleys, which may not represent the spatial variation of precipitation in nearby mountain ranges. The gridded climate products from regional and global data sets provide a good deal of uncertainty due to interpolation approaches and a trade-off between resolution and availability of observed data (Li *et al.*, 2017).

Various hydrological modelling approaches have been set up for several basins of the central Himalayas, at different spatio-temporal scales, from physically-oriented representations of processes, such as TOKAPI by (Pellicciotti *et al.*, 2012) or SWAT by (Bharati *et al.*, 2016), to more conceptual ones, such as SRM by (Immerzeel *et al.*, 2010), GR4J by (Andermann *et al.*, 2012) and (Pokhrel *et al.*, 2014), GR4JSG by (Nepal *et al.*, 2017a), SPHY by (Lutz *et al.*, 2014), HDSM by (Savéan *et al.*, 2015) and J2000 by Nepal *et al.* (2014, 2017b). However, large discrepancies remain in the representation of hydrological processes among several studies at a regional scale stemming from the variation in modelling applications, input data and the processes taken into account.

For instance, for the Dudh Koshi River basin, annual actual evapotranspiration is estimated at $14\%$, $20\%$ and $52\%$, respectively, of annual precipitation by Andermann *et al.* (2012); Nepal *et al.* (2014) and Savéan *et al.* (2015). Estimations of the snow melt contribution to annual stream flow at the outlet of the Dudh Koshi River basin range from $6\%$ (Andermann *et al.*, 2012) to $27\%$ (Nepal *et al.*, 2014); estimations of the glaciar melt contribution to annual stream flow range from $4\%$ (Andermann *et al.*, 2012) to $19\%$ (Lutz *et al.*, 2014). Moreover, estimations of the contribution of underground water to surface flow are still very divergent because of the variation in methodological approaches. The contribution of groundwater flows to annual stream flows is estimated at about $60\%$, $20\%$ and $12\%$, respectively, by Andermann *et al.* (2012); Nepal *et al.* (2014) and Lutz *et al.*



(2014). The variation is mainly due to the conceptualization of groundwater processes in different models, for example J2000 represents two compartments for groundwater storage, whereas SPHY has one and GR4J has a conceptual representation of groundwater.

5    Taking into account this difficult context, the aim of this paper is to compare the hydrological response at the local scale of two models using a different degree of refinement to represent physical processes in sparsely instrumented mountainous catchments. Two small catchments were chosen to represent different climatic and physiographic characteristics of the Central Himalayas: the Kharikhola ($18.2 km^2$) and the Tauche catchment ($4.65 km^2$) which represent respectively middle mountains and headwaters of high mountains of the Nepalese Himalayas.

The hydrological processes of these two catchments are assessed by applying two hydrological models, namely ISBA and J2000, which differ in their conceptualizations. The ISBA surface scheme (Noilhan and Planton, 1989; Noilhan and Mahfouf, 1996) has been applied in mountainous areas (Martin and Etchevers, 2005; Caballero *et al.*, 2007; Lafaysse *et al.*, 2011) to simulate the interaction between the hydrosphere, the biosphere and the atmosphere at various spatial resolutions. In this study, 15  an additional conceptual module is coupled to ISBA to represent the flow routing, which was not originally included in the surface scheme. The J2000 model applies a process-based approach through calibration parameters and is distributed based on Hydrological Response Units (HRUs). The J2000 model is applied in Himalayan catchments but at meso-scale catchments such as the Dudh Koshi river basin. The novelty of the study is the application of a fully distributed surface scheme based on mass and energy balance equations and a semi-distributed calibrated model, at a small spatio-temporal scale, and under the same 20  forcing conditions and parametrization. Uncertainties associated with both climatic variables and static spatial parametrization for topography, soil and vegetation were minimized as much as possible using a similar input data set for two models. In doing so, the uncertainty analysis is focused on the structural uncertainties associated with the structures of the models and their impacts on hydrological simulations. Indeed, the comparison of two models is particularly of benefit to estimate structural uncertainties in the modeling approaches.

25  **1   Study area**

The Kharikhola and Tauche sub-catchments are part of the Dudh Koshi River basin in Eastern Nepal. This basin has a steep topography and high mountain peaks including Mt Everest, (8848, m a.s.l), dominated by a sub-tropical climate in lower areas and an alpine climate in high-altitude areas (see FIGURE 1). These two sub-catchments present different climatic and physiographic characteristics.

30

The elevation of the Kharikhola catchment varies from from 1980 m a.s.l. to 4660 m a.s.l. with an area of 18.20 $km^2$. This catchment is covered by extensive agricultural areas (below 2500 $m.a.s.l$), forests ( between 2500 $m.a.s.l$ and 3500 $m.a.s.l$) and sparce vegetation areas (above 3500 $m.a.s.l$). The glaciar coverage on the Kharikhola catchment is nil. The elevation of



the Tauche catchment varies from from 3980 m a.s.l. to 6110 m a.s.l. with an area of 4.65 $km^2$. This catchment is sparsely vegetated, mainly covered by shrublands or alpine steppes. On the Tauche catchment, the Tauche peak glacier is suspended upstream of the catchment and accounts for about 0.37% of the basin's total area, according to Racoviteanu *et al.* (2013) up-to-date glaciar inventory. The glacial contribution to the flow for the Tauche catchment is therefore considered to be negligible

5 and is not included in the modelling applications. The main morphological characteristics of the two catchments studied are summarized in TABLE 1.



**Figure 1.** Map of the studied area: (A) the Dudh Koshi River basin at the Rabuwabazar gauging station, managed by the Department of Hydrology and Meteorology of the Nepal Government. The (B) Tauche and (C) Kharikhola sub-catchments are defined by the corresponding gauging stations. Source: OpenStreetMaps, photos by Rémi Muller (D) and Judith Eeckman (E).



**Table 1.** Summary of the main morphological characteristics of the two catchments studied: : Kharikhola catchment and Tauche catchment (Nepal), which represents mid-altitude mountains and high-mountain headwaters, respectively.

|  | Kharikhola | Tauche | unit |
| --- | --- | --- | --- |
| Area | 18.2 | 4.65 | $km^2$ |
| Elevation range | 1980 - 4660 | 3980 - 6110 | m.a.s.l. |
| Glaciarized area | 0% | 0.37% | - |
| Discharge data | from 2014-05-03 to 2016-05-20 | from 2014-05-07 to 2016-05-09 |  |

## 2 Modelling approaches

The implementation choices are summarized for both models in TABLE 2.

### 2.1 ISBA and routing module

The ISBA (Interaction Soil Biosphere Atmosphere) surface scheme (Noilhan and Planton, 1989; Noilhan and Mahfouf, 1996) is implemented in the SURFEX platform (Masson *et al.*, 2013) to represent the nature land tile. The ISBA surface scheme simulates vertical fluxes between the soil, vegetation and the atmosphere at a sub-hourly time step (SVAT model). Different implementations of soil transfers, vegetation, sub-grid hydrology and snow processes are available in SURFEX. Implementations of ISBA functions described in TABLE 2 are used in this study. The explicit multilayer version of ISBA (ISBA-DIF) (Boone *et al.*, 2000; Decharme *et al.*, 2011) and the explicit snow scheme in ISBA (ISBA-ES) (Boone and Etchevers, 2001; Decharme *et al.*, 2016) are used in this work. The transport equations for mass and energy are solved using a multilayer vertical discretization of the soil (diffusive approach). The number of layers for the Kharikhola and Tauche catchments is forced to nine layers in order to limit the computation time. In addition, a twelve-layer vertical discretization of snow pack and provides a mass and energy balance for each layer, taking into account snow-melt and snow sublimation. The Horton (Horton, 1933) and Dunne (Dunne, 1983) runoff mechanisms are modeled using a sub-grid parameterization described in Decharme and Douville (2006). ISBA is set up for the Tauche and Kharikhola catchments on a regular grid at a 400-m spatial resolution and with an hourly time step.

Since the dependency between mesh cells is not initially implemented in the SURFEX platform, an additional routing module was implemented and coupled to ISBA offline simulations. This module is adapted from the HDSM (Hydrological Distributed Snow Model) model, implemented and used by Savéan *et al.* (2015). The structure of the module is extensively described in Savéan (2014). For each cell, surface runoff (given by the sum of Dunne runoff and Horton runoff) and the drainage at the bottom of the soil column are directed toward two simple linear reservoirs, $R_s$ and $R_d$ respectively. Residency times in $R_s$ and $R_d$ (respectively, $t_s$ and $t_d$) are calibrated as uniform parameters over the catchment. The sum of the output flows of $R_s$ and $R_d$ is then directed toward the transfer reservoir, which allows propagating the flows according to terrain orography. The residency time in the transfer reservoir is defined for each mesh point as the ratio between the flow velocity and the distance from the



**Table 2.** Summary of ISBA surface scheme and J2000 model structures, for precipitation phase distribution, interception, evapotranspiration, snow accumulation and melt, soil water, runoff components, groundwater and flow routing treatments.

| ISBA | J2000 |
| --- | --- |
| **Precipitation** | |
| *For both models*: Precipitation is distributed between rain and snow according to the same threshold temperatures for both models. | |
| **Interception** | |
| *For both models*: Simple interception storage approach (Dickinson, 1984). The interception storage is computed according to the vegetation type defined by its Leaf Area Index (LAI) for rain and snow. | |
| **Evapotranspiration (ET)** | |
| ET results from the water and energy balance applied on bare soil, vegetation and snow-cover (Noilhan and Planton, 1989). | The potential ET is calculated by Hargreaves and Samani (1982) and is then checked against actual water storage in different land-scape compartments (such as interception, soil water etc) to calculate actual ET. |
| **Snow accumulation and melt** | |
| The ISBA-ES implementation (Boone and Etchevers, 2001; Decharme *et al.*, 2016) provides a twelve-layer discretization of the snow pack. Mass and energy balances are computed for each layer, considering snow-melt and sublimation. | Potential melt from snow pack is estimated with energy input from temperature, rain and ground surface. Accumulation and melting can occur within a time step, controlled by separate accumulation or melt temperatures (Knauf, 1980). |
| **Soil water** | |
| The diffusive approach (ISBA-DIF), (Boone *et al.*, 2000; Decharme *et al.*, 2011) uses a 14 layer discretization of the mixed-form richard's equation with vertcal soil water fluxes represented by Darcy's law. | Middle/large pore storage (MPS/LPS) partition. MPS refers to the field capacity, whereas LPS refers to the flowing water in the soil that generates subsurface runoff and percolation to groundwater reservoirs. |
| **Runoff components** | |
| *For both models*: The notions of Dunne's flow (saturation excess runoff) and Horton's flow (infiltration excess runoff) are considered in the computation of surface runoff. | |
| Dunne's and Horton's runoffs are controlled according to (Dümenil and Todini, 1992). The Dunne runoff for each grid cell depends on the fraction of the cell that is saturated. | Saturation excess runoff and infiltration excess runoff together provide overland flow (RD1) (Krause, 2001, 2002). When LPS is filled, the excess water is divided into sub-surface flow (RD2) and percolation to the groundwater reservoir. |
| **Groundwater** | |
| Groundwater storage is reated by an additional conceptual module. Drainage at the bottom of the soil column is stored in a linear reservoir, controlled by a calibrated residency time. | The percolated water is distributed into two groundwater compartments, which produce interflow 2 (RG1) from shallow aquifers and baseflow (RG2) from deep aquifers. |
| **Routing** | |
| Flow routing is treated by an additional conceptual module. The outflow is computed for each grid cell according to the average slope of the cell, weighted by a calibrated velocity coefficient. | The four different runoff components (RD1, RD2, RG1 and RG2) from each HRU are routed to the next connected HRU until it reaches a river network, using a simplified kinematic wave approach (Krause, 2001). |



centre of the mesh point to the centre of the previous upstream mesh point. The flow velocity is calculated as the ratio of the mesh point slope and a reference slope, taken equal to the catchment median slope. This ratio is weighted by a $c_{vel}$ transfer coefficient. $c_{vel}$ is calibrated as a uniform parameter. The code for this routing module is implemented in fortran90 language and available at *www.papredata.org*.

## 2.2  J2000 modelling system

The J2000 hydrological model is a process-oriented hydrological model (Krause, 2001). The model is implemented in the Jena Adaptable Modelling System (JAMS) framework (Kralisch and Krause, 2006; Kralisch *et al.*, 2007), which is a software framework for component-based development and application of environmental models. The J2000 model includes the main hydrological processes of high-mountain catchments. A short description of the main processes has been provided in TABLE 3. A more detailed description is provided by Krause (2001) and Nepal (2012). The J2000 model has already been applied to Himalayan catchments (Nepal *et al.*, 2014, 2017a).

To optimize the J2000 model parameters for the KhariKhola and Tauche catchments, we used the base parameter set from a previous study by (Nepal *et al.*, 2014), which was defined for the Dudh Koshi River basin at the Rabuwabazaar gauging station (3712 km$^2$). Similarly, (Nepal *et al.*, 2017a) also used the same parameter sets for nearby Tamor sub-catchment (4004 km$^2$) to argue that spatial transferability of the J2000 model parameters is possible in neighbouring catchments with physical and climatic similarities between the catchments. Out of 30 parameters, six parameters were optimized further to match the catchment responses in the KhariKhola and Tauche catchments: the groundwater recession coefficient for baseflow (gwRG2Fact), the coefficient for the distribution of water between the upper and lower zone of groundwater (gwRG1RG2dist), the recession coefficient for RD1 and RD2 (soilConcRD1 and soilConcRD2), maximum percolation (soilMaxPerc) and baseTemp for snowmelt and the parameter to distribute precipitation into rainfall and snow (trs). The recession coefficient for floods from (Nepal *et al.*, 2014) is not applied here because of the local scale catchments. Because of the basin size and climatic variability within the catchment and related scale issues, optimization of parameters is suggested. The description of these parameters along with their dimensions are available in Nepal *et al.* (2017a).

## 2.3  Spatial discretization methods

The SPOT DEM (Gardelle *et al.*, 2012), as well as soil and land cover maps are provided for both catchments at the 40-m resolution. In ISBA, the catchments are discretized over a regular grid at the 400-m resolution. Sixty-nine grid cells are defined for the Kharikhola catchment and 28 grid cells are defined for the Tauche catchment. In J2000, the catchments are discretized into 346 and 132 HRUs, respectively. The minimum size of HRUs is forced to be larger than 5 DEM pixels, i.e. 0.008 $km^2$. TABLE 3 summarizes the results of the spatial discretization for both modelling applications. FIGURE 3 shows the hypsometric information of the land surface area in different elevation zones. Although the overall pattern of hypsometry is similar in both models, they tend to show fairly opposite area coverage above and below about 3000 $m.a.s.l.$ for Kharekhola and 5000 $m.a.s.l.$ for Tauche.





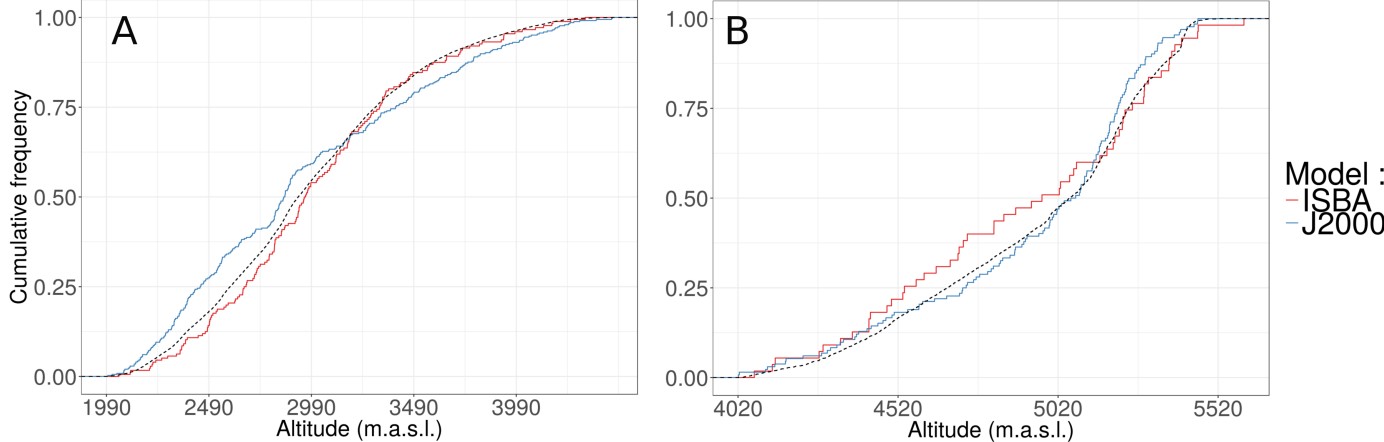

**Figure 2.** Hypsometric curve for Kharikhola catchment (A) and for the Tauche catchment (B), provided by the IBSA discretization on a regular grid at the 400-m resolution (red curve) and by the J000 discretization into HRUs (blue curve). The dotted line is the hypsometric curves given by the 40-m SPOT DEM (Gardelle *et al.*, 2012).

**Table 3.** Summary of the spatial discretization methods used in ISBA and in J2000, for the Kharikhola and Tauche catchments.

|  | Kharikhola catchment | | Tauche catchment | | |
|---|---|---|---|---|---|
|  | ISBA | J2000 | ISBA | J2000 | |
| Number of units | 69 cells | 346 HRUs | 28 cells | 132 HRUs | |
| Minimum size of units | 0.16 | 0.008 | 0.16 | 0.008 | $km^2$ |
| Minimum altitude | 2050 | 1997 | 4070 | 4021 | m.a.s.l. |
| Maximum altitude | 4326 | 4459 | 5600 | 5457 | m.a.s.l. |

## 2.4 Soils and vegetation patterns

The physical characteristics of soils and vegetation are defined in both models using a classification containing nine categories of soil/vegetation entities, defined in the field and extrapolated using a semi-supervised classification of two Sentinel 2 images (Drusch *et al.*, 2012) at a 10-m resolution for the two catchments studied. For each of the nine classes, values for soil depth and texture, root depth, vegetation type and vegetation fraction, leaf index area, surface albedo and surface emissivity are derived both from in-situ measurements and from other available products. The classification method and the characteritics of each class are described in detail by Eeckman *et al.* (2017). The surface classification established at the 10-m resolution is aggregated at the resolution of each model. The classification maps used to parameterize soil and vegetation in both models, for both the Kharikhola and Tauche catchments, are presented in FIGURE 3. The overall location of each class is consistent in both models, although the two different spatial aggregation methods necessarily induce local differences in these maps.





In situ measurements showed that soils are in general poorly developed and mostly sandy (with the sand fraction about 80% and the clay fraction about 1% on average over all the soil samples measured). Soil depths vary from around 1.3 m in cultivated areas to 35 cm for high-mountain steppes.

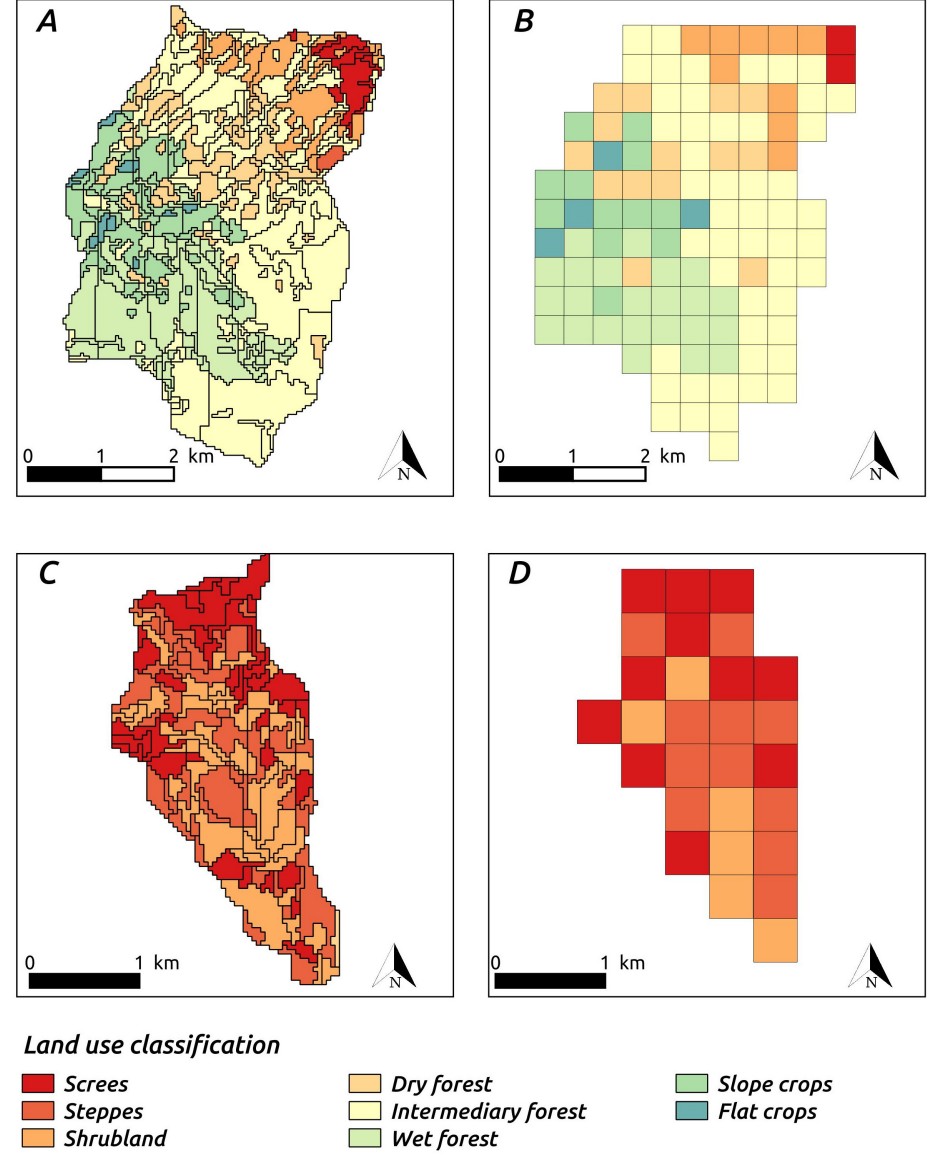

**Figure 3.** Land cover classification defined for each HRU in J2000: (A) on the Kharikhola catchment, (C) on the Tauche catchment; and on a regular 400-m resolution grid in ISBA: (B) on the Kharikhola catchment, (D) on the Tauche catchment. Each land cover class provides soil and vegetation characteristics established from in situ measurements.





## 2.5 Discharges

Hourly discharge time series are available at the hydrometric stations located at the Kharikhola outlet and at the Tauche outlet, from 2014-05-03 to 2016-05-20 and from 2014-05-07 to 2016-05-09, respectively (see TABLE 1). The time series at Kharikhola station contains 34% missing data in 2014-2015, due to damage to the sensor. The time series at Tauche station

contains no missing data, but additional observations made by a local observer indicated that the river was frozen from 2015-01-22 to 2015-02-28 and from 2016-01-08 to 2016-02-23. Discharge is considered as nil during frozen periods.

The Base Flow Index (BFI) (Hingray *et al.*, 2009) was computed for each of the two discharge time series observed to assess the ratio between the volume of the base flow and the total discharge observed, over the period of availability of measurements,

using the lfstat R library (Koffler and Laaha, 2013). The obtained BFI values obtained show that in Kharikhola, 55% of the discharge is provided by base flow and in Tauche, 79% of the discharge is provided by base flow.

## 2.6 Climatic input

Temperature and total precipitation are measured at 11 weather stations installed within the Dudh Koshi basin (see FIGURE 1). Reliable measurements for short- and long-wave radiation, atmospheric pressure, relative air humidity and wind speed are

available at the Pyramid station, located at 5035 m.a.s.l. (see FIGURE 1). Hourly time series are computed from measurements over the three hydrological years 2013-2012, 2014-2015 and 2015-2016. The hydrological year is considered to start on April 1, as decided by the Department of Hydrology and Meteorology of the Nepalese Government and in general use (Nepal *et al.*, 2014; Savéan *et al.*, 2015). Two seasons are defined: the monsoon season, from April 1 to October 30, and the winter season, from November 1 to March 31.

Climatic variables are spatially interpolated according to the methods and values detailed in Eeckman *et al.* (2017):

- Air temperature measurements are spatially interpolated using a multi-linear method weigthed by the inverse distance (IDW method), coupled with a seasonal altitudinal lapse rate. The altitudinal lapse rate is computed from the observation : $-5.87°C.km^{-1}$ for winter and $-5.64°C.km^{-1}$ for monsoon.

- Total precipitation is interpolated using the method proposed by Valery *et al.* (2010): the IDW method is coupled to a multiplicative altitudinal factor $\beta$. The altitudinal factor $\beta$ is represented as a piecewise linear function of altitude. Altitudinal thresholds and lapse rates are optimized to provide optimal bias on annual discharge for both the Kharikhola and Tauche catchments. During the monsoon season, precipitation is considered to increase up to an altitudinal threshold

30    of 3470 m.a.s.l. (3113 m.a.s.l. during winter) at a rate of 0.032 $km^{-1}$ (1.917 $km^{-1}$ during winter), then to decrease at a rate of -1.382 $km^{-1}$ (-1.83 $km^{-1}$ during winter) up to 3709 m.a.s.l. (4943 m.a.s.l. during winter). For higher altitudes,



precipitation is considered to decrease at a rate of -0.283 $km^{-1}$ (-0.191 $km^{-1}$ during winter).

- Long-wave radiation, atmospheric pressure and specific air humidity measurements at the Pyramid station are spatialized as a function of altitude, using the method proposed by Cosgrove *et al.* (2003). Since short-wave radiation and wind speed have a quite low sensivity in the models in comparison with the other variables, these two variables are not spatially interpolated and are considered to be equal to the measurements at the Pyramid station for the two catchments studied.

This interpolation method for precipitation provides optimal precipitation fields for both the Kharikhola and Tauche catchments, for the two hydrological years 2014–2015 and 2015–2016, according to the simulated discharges at the outlet. However, the interannual variability is hardly represented in this interpolated data set. Indeed, these 2 years are very different. For the Kharikhola catchment, observed discharge at the outlet reached 48.3 $mm/day$ in July 2014, whereas it did not exceed 24.5 $mm/day$ in 2015–2016 (see FIGURE 4). For the Tauche catchment, the rainfall-runoff ratio was 53% in 2014–2015 and 82% in 2015–2016, considering interpolated precipitation and observed discharge TABLE 4). These variations can be due to the combined effects of (i) the effective interannual variability of climatic variables, (ii) errors in precipitation measurements, in particular concerning snowfall underestimation (Sevruk *et al.*, 2009), (iii) errors in water level measurements or in the interpolation of discharge based on the rating curve. In particular, high discharge peaks might be overestimated when interpolated from the rating curve, because only a few gauging points are available for high water levels.

However, since the aim of this paper is to compare the hydrological responses of two models when using the same input data set, the choice was made not to consider uncertainties in hydro-climatic input data, but to focus on comparing the simulated responses of the two models.

## 2.7 Snow cover area

The MOD10A2 product (Hall *et al.*, 2002) provides the maximum snow cover extent over a 500-m resolution grid, at an 8-day time scale since 2000-02-26 to the present. MOD10A2 is derived from the MODIS/Terra Snow Cover Daily product (MOD10A1). To compute the MOD10A2 maximum snow cover extent from MOD10A1 snow cover, the following condition is applied: if a pixel if considered as covered by snow at least once within each 8-day time lapse in the MOD10A1 product, this pixel is considered as covered by snow for the corresponding 8-day period in MOD10A2. MOD10A2 is commonly used in glaciological and hydrological studies in the western Himalayas (Shrestha *et al.*, 2011; Panday *et al.*, 2014; Pokhrel *et al.*, 2014; Savéan *et al.*, 2015). Moreover, the accuracy of this product was assessed in mountainous areas by various studies (Jain *et al.*, 2008). In particular, Chelamallu *et al.* (2014) concluded that the MODIS products were more accurate in regions with substantial snow cover than in regions with low snow cover.



## 2.8    Modelling strategies

Observed discharges were available for only 1 complete hydrological year (2015-2016) at the Kharikhola catchment and for
2 hydrological years at the Tauche catchment (2014-2016). The ISBA and J2000 simulations over these catchments were run
separately from 2013-01-01 to 2016-03-31. The 2013–2014 year was used as a spin-up period and the results were observed
for the 2014–2016 hydrological years. The ISBA was run at an hourly time scale and hourly model outputs were aggregated to
the daily level. The J2000 was run at a daily time step. The J2000 model parameters and the ISBA routing module parameters
were calibrated over the whole period of observed discharges available. No independent validation period was then considered
here due to the short period of observed data.

The same land cover and soil types maps have been used to define surface parameters in both models (see Section 2.4). The
same soil depths and textures values have been used to define ISBA soil parameters and maximal volumes for MPS and LPS
reservoirs in J2000 (see Section 2.2). Other parameters in both models are calibrated according to the same discharge data.
This protocol allows to reduce differences in simulation results due to model parameterization in both models. However, since
parameters have different physical meaning in both models, they can not be exactly equal. Thus, the choice has been made in
this work not to consider uncertainties associated with model parametrizations and to focus on uncertainties associated with
model structures.

Model performance was assessed against observed discharge data using the four efficiency criteria : coefficient of deter-
mination $r^2$, Nash-Sutcliffe Efficiency ($NSE$), $NSE$ for the square root of discharges ($NSE_{sqrt}$) and relative bias ($Bias_r$),
computed at the daily time scale. The $NSE_{sqrt}$ has the property of flattening flow peaks and therefore it is used to assess per-
formance for low-flow periods (Krause $et$ $al.$, 2005). To assess performance for high-flow, the $NSE$ criteria is also computed
separately for the high flow periods, i.e. from June 1 to September 30. This criterion is noted $NSE_{high}$.

## 3    Results and discussion

TABLE 4 presents annual volumes for total precipitation, solid precipitation, evapotranspiration, discharge and snow-melt
contribution, in annual average over each of the two catchments studied. FIGURE 4 and FIGURE 5 present the dynamics of
simulated variables in both models, respectively for the Kharikhola and Tauche catchments.

### 3.1    Evaluation against observed discharge

FIGURE 4 and FIGURE 5 show the simulated and observed hydrographs in the Kharikhola and Tauche catchments, respec-
tively. TABLE 4 presents the performance of both models for four different efficiency criteria computed at the daily time scale.





**Table 4.** Annual volumes for input variables (in millimetres per year): total precipitation, solid precipitation and for variables simulated by ISBA and J2000 models: actual evapotranspiration, discharge at the outlet, snow-melt contribution, snow pack storage variation and soil storage variation, for the 2014–2015 and 2015–2016 hydrological years, for the Kharikhola and Tauche catchments. Performance criteria ( Nash-Sutcliffe Efficiency $NSE$, relative bias $Bias_r$, determination of coefficient $r^2$, $NSE$ for the square root of discharges $NSE_{sqrt}$ and $NSE$ computed for the high-flow period $NSE_{high}$), computed at the daily time scale are also provided.

| | Kharikhola catchment | | | | Tauche catchment | | | |
| --- | --- | --- | --- | --- | --- | --- | --- | --- |
| | 2014-2015 | | 2015-2016 | | 2014-2015 | | 2015-2016 | |
| Observed discharges | - | | 1800 | | 440 | | 477 | |
| Model | ISBA | J2000 | ISBA | J2000 | ISBA | J2000 | ISBA | J2000 |
| Total precipitation | 3034 | 3064 | 2256 | 2254 | 837 | 824 | 581 | 607 |
| Solid precipitation | 42 | 36 | 27 | 26 | 403 | 281 | 245 | 148 |
| Actual evapotranspiration | 579 | 548 | 622 | 555 | 292 | 372 | 285 | 363 |
| Discharge at the outlet | 2346 | 2523 | 1631 | 1803 | 373 | 413 | 385 | 303 |
| Snow-melt contribution | 53 | 50 | 27 | 21 | 336 | 276 | 309 | 199 |
| Snow pack storage variation | 0 | 0 | 0 | 0 | -66 | -26 | 66 | 24 |
| Soil storage variation | -41 | -16 | 33 | 15 | -8 | -7 | 7 | 6 |
| $NSE$ | 0.5018 | 0.60453 | 0.9010 | 0.9158 | 0.8958 | 0.9194 | 0.6760 | 0.5172 |
| $Bias_r$ | -45.7 | -37.1 | -9.7 | 0.04 | -11.7 | -2.8 | -19.5 | -39.0 |
| $r^2$ | 0.8613 | 0.9049 | 0.9120 | 0.9327 | 0.9352 | 0.9453 | 0.7203 | 0.7944 |
| $NSE_{sqrt}$ | 0.6645 | 0.6985 | 0.8733 | 0.9395 | 0.8553 | 0.8219 | 0.6888 | 0.6956 |
| $NSE_{high}$ | 0.0742 | 0.1512 | 0.7629 | 0.6640 | 0.7329 | 0.8400 | 0.0193 | -0.7239 |

Annual relative bias on discharge for the Kharikhola catchment is satisfactory for 2015–2016 for both models (-9.7% for ISBA and 0.04% for J2000), but the discharge at the Kharikhola outlet is strongly under-estimated for 2014–2015 for both models ($Bias_r$ is -45.7% for ISBA and -37.1% for J2000). On the Tauche catchment, the observed discharges are under-estimated for both years, for both models, with average $Bias_r$ values of -15.6% for ISBA and -20.90% for J2000. These under-estimations are due to the under-estimation of total precipitation for the corresponding years, as presented Section 2.6.

For the two hydrological years, the dynamics of the observed discharges is accurately represented by the two models for the two catchments, with the annual average of $r^2$ greater than 0.72 and $NSE$ values greater than 0.70. During the monsoon period, the discharge dynamics is driven by precipitation, with a quick response of the surface runoff for both catchments.





These quick flow variations are satisfactorily represented by both models.

Low flows including rising and recession periods are accurately captured for both years by both models for the Kharikhola catchment: on average over the 2 years, $NSE_{sqrt}$ is 0.77 for IBSA and 0.82 for J2000. For the Tauche catchment, low flows are clearly represented for 2014–2015 ($NSE_{sqrt}$ is 0.78 for ISBA and 0.76 for J2000). However, in 2015–2016, the observed discharge during the recession period (when the river is not frozen) is under-estimated by both models.

The representation of high-flow peaks in the monsoon period is very satisfactory for the Kharikhola catchment in 2015–2016 and for the Tauche catchment in 2014–2015 ($NSE_{high}$ values greater than 0.66). However, flow peaks are strongly under-estimated for the Kharikhola catchment in 2014–2015 and for the Tauche catchment in 2015–2016, with low $NSE_{high}$ values. The simulation of particular flood events will be further discussed in section 3.3.

## 3.2 Components of annual water budgets

### 3.2.1 Precipitation

Slight differences exist for total precipitation between the J2000 and ISBA models, although the same precipiation input is provided for both models. These differences stem from both the spatial discretization methods used in both models for precipitation spatial interpolation (see section 2.3). Indeed, even though the input grid data provided for precipitation are the same for both models, precipitation is further interpolated by J2000 from the grid scale to the HRU scale, using the inverse distance weighting method. However, for both catchments,the difference in total precipitation between the two models represents less than 1% of the annual volume (0.98% for the Kharikhola catchment and 0.45% for the Tauche catchment). The difference between both models for total precipitation can then be considered as negligible.

In addition, for the Kharikhola catchment, the difference of annual average solid precipitation is also considered as negligible (7 $mm$, representing 1.1% of annual average solid precipitation). For the Tauche catchment, average solid precipitation is 219 $mm$ higher for ISBA than for J2000. This difference represents about 40% of the annual volumes of solid precipitation. This significant difference is due to i) the difference in the spatial discretization methods used in both models 2) the difference in the time step used for precipitation phase distribution in both models. Indeed, precipitation phase in J2000 is computed at the daily time step, whereas it is computed at the hourly time step in ISBA. The infra-daily variations of solid precipitation is then missed in J2000.

### 3.2.2 Snow-melt contribution

For both models, the contribution of snow-melt to discharge is less than 1.5% for the Kharikhola catchment. For the Tauche catchment, the contribution of snow-melt accounts for 45.3% of the annual simulated discharge in ISBA results, and 33.2% of



the annual simulated discharge in J2000 results.

For the Tauche catchment (see FIGURE 5), both models provide the majority (73% in ISBA and 82% in J2000) of snow-melt during the monsoon season. Amplitude and timing of snow-melt between July and November (monsoon and post-monsoon periods) are similar in both models. However, snow-melt occurring before monsoon periods in 2014 and 2015 is greater in J2000 than in ISBA. Minimum air temperature for snow-melt to occur is -0.9 $°C$ in J2000 and -9.8 $°C$ in ISBA. In ISBA, snow-melt at such a low temperature occurs mainly during winter. This process can be explained by the influence of high solar radiation and important wind velocities at such high elevations, together with low solid precipitation intensities (less than 1 $mm$ per day on average). Moreover, for ISBA, the infra-daily variations of the air temperature significantly influence snow-melt.

### 3.2.3 Evapotranspiration

On average over the 2 hydrological years, estimation of annual actual evapotranspiration (actET) on the Kharikhola catchment was 22.6% of total annual precipitation with ISBA and 19.8% with J2000. On the Tauche catchment, it was 34.4% with ISBA and 50.6% with J2000 of the total annual precipitation. These values include bare soil evaporation, vegetation transpiration and snow sublimation. ActET for the two models in both catchments correlated acceptably at the daily time scale, with $r2 = 0.48$ for the Kharikhola catchment and $r2 = 0.38$ for the Tauche catchment. However, a major difference can be seen in the pre-monsoon period (March-June) in the Tauche catchment where actET from J2000 is higher than ISBA. This period of discrepancy between the two models will be further discussed in section 3.3.

### 3.2.4 Soil water content

The conceptualizations of the soil water storage in both models are very different (see TABLE 2). Considering these stuctural discrepancies, the total water content of the soil column simulated in ISBA can be compared to the sum of the volumes stored in MPS and LPS reservoirs in J2000.

On the Kharikhola catchment, the dynamic of the soil water content of the two models matches well during the high-flow periods (between June and October). During the low-flow periods, the soil water content simulated in ISBA is lower than in J2000. On the Tauche catchment, even though the dynamics of the simulated soil water content are also similar in the two models, the soil water content is permanently higher in J2000 than in ISBA. On average over the two hydrogical years, the variation of soil water content is low for both models and both catchments (see TABLE 4).

In ISBA, the simulated soil water content is limited by the soil humidity at saturation ($w_{sat}$, in $mm$). $w_{sat}$ values are calculated according to Clapp and Hornberger (1978), as a function of soil texture. In J2000, the volume stored in each reservoir MPS and LPS is limited by maximum volumes $maxMPS$ and $maxLPS$, respectively. $maxMPS$ and $maxLPS$ are computed according to soil texture for each HRU. The TABLE 5 presents the average values of $w_{sat}$, $maxMPS$ and $maxLPS$





for the Kharikhola and Tauche catchment. Provided value for $w_{sat}$ is greater than the sum $maxMPS + maxLPS$ for the Kharikhola catchment, but it is lower for the Tauche catchment. This parametrization can explain that the soil water content is globally higher in ISBA than in J2000 for the Kharikhola catchment, but lower in ISBA than in J2000 the the Tauche catchment.

**Table 5.** Parametrisation of soil water content introduced in ISBA and in J2000. $SAND$ and $CLAY$ are respectively the average sand and clay fractions of the soil for each catchment. $w_{sat}$ is the water content of the soil column at saturation computed in ISBA. $maxMPS$ and $maxLPS$ are the maximal storage capacity in MPS and LPS reservoirs in J2000.

| Catchment | SAND | CLAY | $w_{sat}$ | $maxMPS$ | $maxLPS$ |
| --- | --- | --- | --- | --- | --- |
| | | | $mm$ | $mm$ | $mm$ |
| Kharikhola | 79.9% | 1.1% | 207 | 98 | 100 |
| Tauche | 80.9% | 1.7% | 52 | 35 | 32 |

### 3.2.5 Discharge components

The respective contribution of surface overland flow and drainage (i.e. flow in the soil) to discharges at the outlet can be compared in both models. For both models, the surface overland flow is considered as the sum of the Dunne runoff and Horton runoff. The drainage flow at the bottom of the soil column in ISBA is comparable to the sum of the three flows in the soil in J2000 ($RD2$, $RG1$ and $RG2$). The following comparison is given on average over the two hydrological years studied.

For ISBA, the Hortonian runoff represents less than 1% of discharge on the Tauche catchment and about 5% of discharge on the Kharikhola catchment. While in ISBA, this means than the surface flows occur mainly (on the Kharikhola catchment) or almost only (on the Tauche catchment) by the saturation of soil reservoirs rather than by excess infiltration capacity.

For the Kharikhola catchment, the contribution of drainage to discharge is 77% for ISBA and 87% for J2000. For the Tauche catchment, this contribution is 70% for ISBA and 85% for J2000. The overland flow contribution to discharge is 30% for ISBA and 13% for J2000 on the Kharikhola catchment. For the Tauche catchment, this contribution is 23% for ISBA and 15% for J2000. These figures highlight the significant contribution of soil water to discharge for both middle- and high-mountain catchments.

These fractions can be compared to the Base Flow Index (BFI, see Section 2.5) values presented in the data section. For the Kharikhola catchment, the base flow contribution to discharge seems to be over-estimated compared to the BFI values computed on observed discharges (55%). For the Tauche catchment, the BFI values computed on observed discharges (79%) is between the values provided by the two models.





Therefore, this model intercomparison reveals that most discharge at the outlet is provided by drainage. This result is consistent with the description of soils for the two catchments: sandy soils allow fast infiltration, resulting in a larger fraction if the flow occuring in the soils than on the surface. However, the definition of drainage strongly differs between the two models. Indeed, in ISBA, the drainage represents the vertical flow at the bottom of the soil column (without routing nor delays), whereas in J2000, it represents the sum of the outflows from the soil water module.

### 3.3 Analysis of particular events

It is difficult to explain the variations of the observed discharge in June 2015 in Tauche. The observed hydrograph increased from 0.06 $m^3/sec$ on 22 June to 0.3 $m^3/sec$ on 26 June (the highest peak of 2015). The precipitation is below 7 mm and remains throughout the period (the discharge event may not be due to precipitation events). The maximum temperature increased from $7°C$ to $8°C$ from 22 to 24 June and then decreased to $5.5°C$ in 25 June. The discharge event may then be due to either snow-melt fluxes or instrument error. This indicates that the high-mountain hydrological processes are complex in nature, mainly due to i) unknown processes such as sudden fluxes from storage (snowpacks or depression), which are difficult to capture through modelling applications, and ii) possible instrument errors (but we could not verify this independently). The latter could be common in high-altitude areas resulting from low temperature (e.g. sensor freezing) and human error.

An interesting period of discordance between the two models occurred between March 2015 and June 2015 (pre-monsoon period) on the Tauche catchment. During this period, simulated actET is higher in J2000 (up to 2 $mm/day$) than in ISBA (less than 0.5 $mm/day$). This delay in the increasing of actET in ISBA is due to late simulated snow-melt in ISBA. Indeed, the simulated snow pack, that limits the evaporation simulated over bare ground, remains in ISBA until June 2015, whereas it melts from March 2015 in J2000.

### 3.4 Comparison with MOD10A2 maximum snow extent

The simuled snow cover area (SCA) is computed by applying a threshold condition on the simulated snow depth (in ISBA) or on the simulated snow water equivalent (in J2000). For each time step (hourly time step in ISBA, daily time step in J2000), each unit of the model (grid cell for ISBA, HRU for J2000) is considered as covered by snow if the snow depth is greater than 60 $mm$ (in ISBA) or if the snow water equivalent is greater than 40 $mm$ (for J2000). These values are consistent with values used by Biskop *et al.* (2016) on the Tibetan Plateau and by Gascoin *et al.* (2015) in the Pyrenees.

Daily simulated SCA and MOD10A2 maximum snow extent is compared in FIGURE 6, on spatial average over the Tauche catchment, for the 2 hydrological years 2014–2015 and 2015–2016. The dynamics of the MOD10A2 SCA is satisfactorily reproduced by both models: a significant snow period occurred between December 2014 and June 2015 and no significant snow pack was simulted or observed between July 2015 and March 2016. However, the snow pack accumulation conditions in J2000 missed the short-duration peaks in J2000, overall, whereas they were simulated by ISBA (but underestimated by about one-third). Snow-melt occurring between March 2015 and May 2015 was faster in J2000 than in ISBA, in particular with an

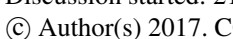



ISBA-simulated SCA limited to around 0.5% over 2 months.

The over-estimation of the SCA by ISBA can be explained by the fact that the land-aspect is not parametrized in ISBA, despite it can significantly influence the snow pack simulation for such contrasted relief. However, since MOD10A2 accuracy at this scale remains to be evaluated (Savéan, 2014), further interperations for model performance ranking according to the MOD10A2 SCA would not be suitable in this study.

## 4    Conclusion

This paper aimed to assess the reliability of the ISBA and J2000 modelling applications to the complex and contrasted environment of high mountain, within small headwater catchments. This paper presents a new conceptual module for soil and surface flow routing, coupled to ISBA to represent lateral transfers. This case study is also, to the author's knowledge, the first attempt to apply the J2000 model at such a high spatial resolution in mountainous areas. In this framework, several points should be underlined:

1. Since empirical models rely more on calibration data, the reliability of a calibrated approach was tested by comparing it to a non-calibrated approach in an environment where data quality and quantity is relatively low. One of the main results of this study is that both models describe the processes for evaporation, quick runoff and discharge in a similar way. The reliability of simulations for these variables can therefore be considered as satisfactory.

2. The differences of structure and the results between the two models concern mainly water storage and flows in the soil, in particular for the storage catchment. Indeed, the representation of soils and underground flows strongly differs between the two models. The consequence is a relatively weak reliability in this compartment of the water budget.

3. The time step used in each model (daily time step in J2000 and hourly time step in ISBA) strongly influences the precipitation phase partition and the snow-melt contribution to discharge. However, for both models, the simulated snow-melt contribution to discharge is negligible for the catchment located in a middle mountain environment, whereas it represents up to 45% of the annual discharge for the catchment located in a high mountain environment.

4. However, the succession of only two climatically very different years does not allow making a definite conclusion on comparing the performance of the two models. Since no additional data is available for independent validation, a formal ranking of the two models is premature. A more complete assessment of uncertainty associated with model structure would include more data sets and would test other complementary model structures.

5. Finally, the results highlight the combined effects of uncertainties associated with (i) the observation and input data, (ii) model parametrization and (iii) the modelling structures. These three sources of uncertainty are necessarily interdependent and a more complete uncertainty analysis would include both sources of uncertainty.



Based on this research, both models present an acceptable reliability in middle- as well as in high-mountain environments. They could be used for operational purposes in two complementary ways: (i) the assessment of water availability considering new scenarios of climate forcing or land use and land cover change and (ii) the sizing of hydraulic installations for agriculture, domestic water supply or hydropower, on the request of the local water users.

**Acknowledgments**

The authors address special thanks to Professor Isabelle Sacareau (Passages Laboratory of the CNRS and Montaigne University of Bordeaux, France), coordinator of the PRESHINE Project. They are also grateful to the hydrometry team and the administrative staff of the Laboratoire Hydrosciences Montpellier, France, the hydrologists and glaciologists of the Institut

des Geosciences de l'Environnement in Grenoble, France, the meteorologists of the Centre National de la Recherche Meteorologique in Toulouse and Grenoble, France, the Association Ev-K2 CNR and the Pyramid Laboratory staff in Bergamo, Italy, Kathmandu and Lobuche, Nepal as well as the Vice-Chancellor of the Nepalese Academy of Science and Technology (NAST) and its staff, especially Devesh Koirala and Anjana Giri. The views and interpretations in this publication are those of the authors and are not necessarily attributable to their institutions.

**Funding**

This work was funded by the Agence Nationale de la Recherche (references ANR-09-CEP-0005-04/PAPRIKA and ANR-13-SENV-0005-03/PRESHINE), Paris, France. It was locally approved by the Bilateral Technical Committee of the Ev-K2-CNR Association (Italy) and the NAST within the Ev-K2-CNR/NAST Joint Research Project. It is supported by the Department of Hydrology and Meteorology, Government of Nepal. The study was also supported in part by ICIMOD core funds contributed

by the Governments of Afghanistan, Australia, Austria, Bangladesh, Bhutan, China, India, Myanmar, Nepal, Norway, Pakistan, Switzerland, and the United Kingdom.





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





**Figure 4.** Daily time series for input variables : total precipitation (PTOT), solid precipitation (SNOWF) and air temperature (TAIR) and for variables simulated by ISBA and J2000 models at the daily time scale : discharge at the outlet (DISCHARGE), actual evapotranspiration (EVAP), soil water content (WGTOT), snow water equivalent (WSN) of the snow pack and snow-melt (MLT), for the 2014–2015 and 2015–2016 hydrological years, for the Kharikhola catchment. Black line is the daily observed discharge at the outlet.



**Figure 5.** Daily time series for input variables : total precipitation (PTOT), solid precipitation (SNOWF) and air temperature (TAIR) and for variables simulated by ISBA and J2000 models at the daily time scale : discharge at the outlet (DISCHARGE), actual evapotranspiration (EVAP), soil water content (WGTOT), snow water equivalent (WSN) of the snow pack and snow-melt (MLT), for the 2014–2015 and 2015–2016 hydrological years, for the Tauche catchment. Black line is the daily observed discharge at the outlet.





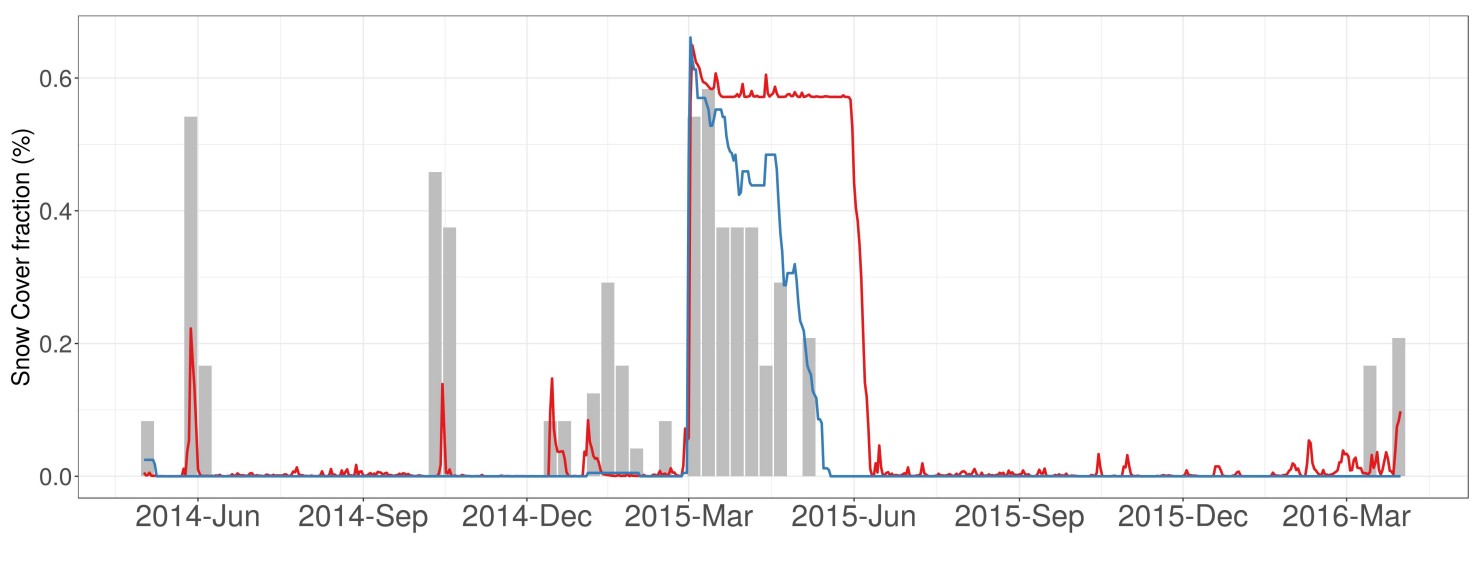

**Figure 6.** Snow cover area simulated by ISBA and J2000, aggregated at the daily time scale, on average over the Tauche catchment, for the 2014–2015 and 2015–2016 hydrological years. Grey bars are MOD10A2 maximal snow cover extend, on average over the Tauche catchment, at a 8 days time scale.