# Peer review of "Assessing reliability of hydrological simulations through model intercomparison at the local scale in the Everest region."

_Hydrology and Earth System Sciences, 2017_

## Referee Comment (RC1) · Anonymous Referee #1 · 19 Sep 2017

The manuscript by Eeckman et al., submitted to HESS for consideration for publication, compares two hydrological models in two very small catchments in the Everest region of the Nepal Himalayas. The high mountain water cycle, especially in the Himalayas, is not well understood and possible consequences of climate change on the water availability in these regions require a thorough understanding of the acting processes. These regions are typically sparsely covered by on-site observations and the quality of the data is often rather poor, deeming scientists to relay on modelling and remotes sensing observations.

This manuscript compares two models, 1) the modified ISBA surface scheme and 2)

[Figure]

the J2000 model. Both models have different parametrizations but can basically run with similar input forcing. Following the description in the manuscript, both models do not consider a deep groundwater compartment, nor do they consider any preferential drainage, e.g. by fractures or discontinuities, in the subsurface. The authors have chosen two very small catchments ($\sim$ 5 and 18 km2) where some onside measurements for river discharge and rain measurements exist. One catchment is situated in the central lesser Himalayas and is covered by five distributed rain gauges as well as one stream gauge at the outlet. The second catchment is situated in the very high Himalayas (no glaciers) and is represented by one rain gauge and one stream gauge, both located at the outlet. Unfortunately, very little information is provided about the data origin and its quality as well as its recording. Evapotranspiration, one of the very under-studied parameters in Himalayan water cycle, was here estimated using ground observations from a distant observatory located at the Everest base camp. The results presented here are basically limited to a comparison of the two models runs and to the hydrograph (without discussing the quality of the local measurements). As the authors point out in the conclusions the models perform well during monsoon, when the hydrograph basically mimics the rainfall pattern, while the long recessions during the post monsoon season have a notoriously weak reliability (with respect to the hydrograph). However, this season is the key to untangle the contributions from the different compartments. This weak performance outside monsoon season indicates that the models are probably not well adapted and eventually miss out important settings. At the same time, the periods outside of monsoon are the periods where water availability is most crucial and which might be most affected by a changing climate.

The study by Eeckman et al. presents a hydrological modelling exercise, however, considering the shortcomings in the models design and the relative spars equipped training catchments I think the manuscript is better suited for publication in a more modelling oriented journal. The main points of criticism are as follow: 1) the results do not contribute to understand the process of the Himalayan water cycle; 2) the input parameters of the models are not rigorously presented and the on-site monitoring

network, especially in the high Tauche catchment, is not well designed as test ground for model performance; 3) and last, I doubt that the two very small catchments can be considered as representative for any of the Himalayan geographical units and thus the findings are difficult to generalize in order to provide modelling advice to local decision makers or scientist around the world.

General comments:

* The authors criticize that most of the stations in high mountains are located low in the landscape. From the maps in figure 1, it seems that all the used stations are located along rivers and do not cover the high ridges.

* The explanation why the authors have chosen ISBA and J2000 is not entirely clear. Which other models are available and have been already applied successfully in the Himalayan region?

* One of the co-author, S. Nepal, has already published hydrological modelling results for the entire Dudh Koshi catchment (Nepal et al. 2014). How did that model perform for the two small catchments of this study and can the results be compared? Can the model by Nepal et al. 2014 be improved from the small scale findings of this work? My understanding was that the study from 2014 worked already quite well for a much larger region. Furthermore, how do the results of this modelling work compare to the modelling by Savéan et al. 2015, which is also covering the entire Dudh Koshi river catchment?

Specific comments:

* Acronym ISBA has never been explained (Interaction Sol-Biosphère-Atmosphère).

* Figure 1: maps have no geographic coordinates. Looking at figure 1D, do the authors expect that soil water storage can be an important reservoir in such a landscape?

* Table 2: ET, why did the authors decide to use the method by Hargreaves and Samani 1982 and not the empirical elevation method developed for Nepal by Lambert and

Chitrakar 1989 (Mountain Research and Development)?

* Page 10: The discharge data is not well explained. How has the data been obtained? What are the sensors? How has stage height been converted into discharge?

* Page 10: As for discharge, where is the data coming from? How was it recorded? . . .

* Page 10/line 14: I am surprised by the temporal definition of the monsoon. Usually April to beginning of June is termed pre-monsoon and is characterized by snow melting at high elevations. The DHM normally expects the start of monsoon around the 10-15th of June.

* Page 10: The precipitation interpolation method IDW needs to be better explained, to be understandable without reading secondary literature, at least the specific techniques applied in this work.

* Page 11: Please give more details on the method to spatialize radiation, pressure, humidity. . . For the non-Nepal Himalayas expert audience, a more detailed explanation of the Pyramid station is appropriate.

* Page 14: Can groundwater explain the mismatch during low flow?

* Page 15: Please discuss the soil water storage differences between the two models in more details.

* Page 16: It would be helpful if Dunne and Horton runoff are defined somewhere in the manuscript. Especially the differences between the two and how the two models treat these two components.

* Page 16: The authors find that most of the water drains through the soils rather than along the surface. Is this something other studies have already documented? What are the consequences of a preferential drainage through the soil compartment?

* Page 16 line 15: Please explain what you mean by "contribution of drainage to discharge".

* Page 17: How do the snow water equivalent and snow cover findings compare to studies from the Himalayas, e.g. Wulf et al. 2016 (Advances in Water Resources) or Putkonen 2004 (Arctic, Antarctic, and Alpine Research) or the two publications of the Dudh Koshi hydrology Nepal et al. and Savéan et al.?

* Page 18: The authors claim that for the first time they apply the models at such a high spatial and temporal resolution in mountains. Can the authors explain what is the gain of such highly-resolved modelling, especially considering that the input data is much coarser or even from different locations? Has it been tested if such a high resolution is needed or would a rather coarse resolution provide similar results. In that light, what are the errors that are propagated from interpolation of the input data through the model into the results?

* Page18: If the time resolution has such an impact on the precipitation phase partition but the aim was to run both models with the same input parameters, why has the same temporal resolution not been used for both models?

* Page 19: In the final lines the authors state that the models can be used to predict water availability for power generation as well as under changing climate conditions. Can this conclusion really be drawn from the two very small scale studies? How can the models be scaled to different regions? What is the minimum input information needed to obtain quality results?

* Figure 5: Why is there so sharp steps in the measured discharge curve during low flow? And secondly, did it not snow in winter 2015/2016?

---

## Referee Comment (RC2) · Anonymous Referee #2 · 27 Sep 2017

In this paper, submitted by Eeckman et al., two hydrological models are implemented on two small Nepalese basins located in the Himalayas mountain range, with the objective of comparing their simulated responses. The study is challenged by heterogeneous and sparsely instrumented basins, by the short duration of the observation time series, and by the absence of validation data in addition to streamflow. No validation of the models is performed.

General comment:

This paper suffers from the short span of the observations series, preventing any validation that would confirm the generalisation of the optimized parameters and limiting

the analysis to two hydrological years. In the Kharikhola basin, both models have difficulties with the rainier 2014-2015 year, probably because there are many missing streamflow observations that may have helped identifying parameters more suitable to such situation. In the Tauche basin, the models somehow disagree in 2015-2016, which is dryer that the preceding one. From an operational perspective, it is normal that a model does not behave as expected all the time (and there are many reasons for it), but in the present situation the short duration prevents any generalization that may be useful to a general audience. Overall, the paper should probably be targeted towards a more regional audience, since no methodological advancements are proposed.

Major comments:

The issue of the "extreme climate heterogeneities" is central to the justification of this paper but is not documented by the authors: only basin-averaged precipitation rates are provided (Figs 4 and 5). Since the Kharikhola precipitation are captured at five locations, the authors could have explored its heterogeneity, especially since the repetitiveness of the precipitation information for that basin is questioned by the authors in page 13.

In page 3, the authors wrote that "the comparison of two models is particularly of benefit to estimate structural uncertainties in the modeling approaches". I am not convinced that comparing the annual volumes (Table 4) and the daily time series (Figs 4 and 5) of simulated variables is enough to tackle the issue of structural uncertainty. Without longer time series and verification observations in addition to streamflow, the authors are limited to identifying similitudes and differences in both model simulations.

The Base Flow Index is at best an empirical tool. I am surprised that it is used here to evaluate the quality of a land surface model (ISBA) that should reflect physical processes (page 16). I am not much more convinced that it is a good idea for the J2000 model. This needs much further justifications, including verifications in heterogeneous basins such as the Himalayas.

[Figure]

It is important to know which score was used in the calibration process of both models – I guess that it is probably one of the five ones used for verification. The list of selected scores could also be improved. First, r2 is not really a performance indicator and may be removed from the paper. Second, NSEsqrt is an all-purpose score without too much emphasis on low or high streamflows (Oudin et al., 2006). It does not reflect low flow performance as written in the paper. It is NSEinv that is the best option for low flow applications (Pushpalatha et al., 2012). I suggest that the authors consider it as well. Third, NSEhigh is much less common and possibly risky with short duration time series.

Minor comments:

No justifications are provided for the model selection.

The authors should clarify what they meant when writing that "a local observer indicated that the river was frozen". Was it frozen from top to bottom? Air temperature does not seem quite cold enough for it to happen. Otherwise, why impose zero flow during that time?

References:

Oudin, L., Andréassian, V., Mathevet, T., Perrin, C., Michel, C., 2006. Dynamic averaging of rainfall-runoff model simulations from complementary model parameterizations. Water Resour. Res. 42, 2005WR004636.

Pushpalatha, R., Perrin, C., Le, N., Andréassian, V., 2012. A review of efficiency criteria suitable for evaluating low-flow simulations. J. Hydrol. 420–421, 171–182.

---

## Author Comment (AC1) · 20 Oct 2017

**Author's response to Review2**

The review is indicated in italic letters and the corresponding answer is given just below.

**Major comments:**

[Figure]

- *The issue of the "extreme climate heterogeneities" is central to the justification of this paper but is not documented by the authors: only basin-averaged precipitation rates are provided (Figs 4 and 5). Since the Kharikhola precipitation are captured at five locations, the authors could have explored its heterogeneity, especially since the repetitiveness of the precipitation information for that basin is questioned by the authors in page 13.*

The spatial distribution of precipitation and temperature for the two catchments has been extensively explored, based on the observation at the available stations. The method developped to generate precipitation and temperature fields is described in Eeckman et al., 2017 (Providing a non-deterministic representation of spatial variability of precipitation in the Everest region). Not to repeat material from this previous study, as required by the editor's first review, this method has not been repeated in the text.

- *In page 3, the authors wrote that "the comparison of two models is particularly of benefit to estimate structural uncertainties in the modeling approaches". I am not convinced that comparing the annual volumes (Table 4) and the daily time series (Figs 4 and 5) of simulated variables is enough to tackle the issue of structural uncertainty. Without longer time series and verification observations in addition to streamflow, the authors are limited to identifying similitudes and differences in both model simulations.*

The authors agree with this comment. The sentence 'A more complete assessment of uncertainty associated with model structure would include more data sets and would test other complementary model structures.' (p18-l26) precises that structural uncertainties can not be quantified extensively in this work. The

comparison of the two modelling approaches is used to highlight close results between the two models (i.e. periods and/or variables for which robustness of the results is fair) and discordant results (i.e. periods and/or variables for which robustness of the results is weak).

- *The Base Flow Index is at best an empirical tool. I am surprised that it is used here to evaluate the quality of a land surface model (ISBA) that should reflect physical processes (page 16). I am not much more convinced that it is a good idea for the J2000 model. This needs much further justifications, including verifications in heterogeneous basins such as the Himalayas.*

The Base Flow Index (BFI) is used in this work as an additionnal informative criteria. The authors agree that this empirical method can not be used as a validation criteria. Consequenly, the reference to the BFI (p10-L8-11 and p16-L21-25) will be removed without any consequence on the paper.

- *It is important to know which score was used in the calibration process of both models – I guess that it is probably one of the five ones used for verification. The list of selected scores could also be improved. First, r2 is not really a performance indicator and may be removed from the paper. Second, NSEsqrt is an all-purpose score without too much emphasis on low or high streamflows (Oudin et al., 2006). It does not reflect low flow performance as written in the paper. It is NSEinv that is the best option for low flow applications (Pushpalatha et al., 2012). I suggest that the authors consider it as well. Third, NSEhigh is much less common and possibly risky with short duration time series.*

For the routing module coupled to ISBA, the three global parameters are

calibrated against the observed discharge using the three criterias : NSE, the relative bais and the NSE of the root square of the discharge, computed at the daily time step. This sentence will be added p4l3. The optimization method used for the J2000 model is extensively presented in Nepal et al., 2014 and is not repeated in this paper. The NSEinv criteria will be computed for both catchment and will be added in the text to the other criterias used.

**Minor comments:**

- *No justifications are provided for the model selection.*

A review of other modelling approaches recently applied in the central Himalayas is presented in the introduction, from p2l20 to p3l4. In particular, the ISBA and the J2000 models have been previously applied in this region by the authors (see Nepal et al., 2014 and Eeckmanet al., 2017). The ISBA approach represents physical processes and do not rely on validation data. However, this approach requires a important amount of data to physically characterize the environment. On the contrary, the J2000 approache requires only a few physical knowledge of the environment but it relies on validation data. In Himlalaya region, on the one hand, a few characterization of the soil and vegetation and on the other hand, validation data (mainly discharge) are associated with important uncertainties. Consequenly, the question to know whether a calibrated model performs better than a non calibrated model has not been answered in other studies. This paper presents a case study in order to clarify this issue.

- *The authors should clarify what they meant when writing that "a local observer indicated that the river was frozen". Was it frozen from top to bottom? Air*

*temperature does not seem quite cold enough for it to happen. Otherwise, why impose zero flow during that time?*

The local observer indicated that the flow was frozen at the point of the water level measurement. No flow was visible from the outside. We considered that this observation means a null flow at the outlet of the catchment. However, it is indeed possible that a very small flow still remains, especially under the surface ice layer. In this case, a strategy could be not to consider these periods in the efficiency criterias.

---

## Author Comment (AC2) · 20 Oct 2017

**Author's response to Review1**

The review is indicated in italic letters and the corresponding answer is given just after.

**General comments:**

- *The authors criticize that most of the stations in high mountains are located low in the landscape. From the maps in figure 1, it seems that all the used stations are located along rivers and do not cover the high ridges.*

The altitude of the stations ranges between 2078 m and 5035 m. Stations are spread in the catchment slopes, in particular in the Kharikhola catchment. They are also located on variously oriented slopes. No station is indeed installed on ridges or summits, first for physical reasons and secondly because ridges and summits are particularly submitted to site effects.

- *The explanation why the authors have chosen ISBA and J2000 is not entirely clear. Which other models are available and have been already applied successfully in the Himalayan region?*

A review of other modelling approaches recently applied in the central Himalayas is presented in the introduction, from p2-l20 to p3-l4. In particular, the ISBA and the J2000 models have been previously applied in this region by the authors (see Nepal et al., 2014 and Eeckman et al., 2017). The ISBA approach represents physical processes and do not rely on validation data. However, this approach requires a important amount of data to characterize the environment. On the contrary, the J2000 approach requires only a few physical knowledge of the environment but it relies on validation data. In Himalaya region, on the one hand, the soil and vegetation behaviors are poorly available and, on the other hand, validation data (mainly discharge) are highly uncertain. Therefore, the question to know whether a calibrated model performs better than a non calibrated model has not been answered in other studies. This paper presents a case study in order to clarify this issue.

- *One of the co-author, S. Nepal, has already published hydrological modelling results for the entire Dudh Koshi catchment (Nepal et al. 2014). How did that model perform for the two small catchments of this study and can the results be compared? Can the model by Nepal et al. 2014 be improved from the small scale findings of this work? My understanding was that the study from 2014 worked already quite well for a much larger region. Furthermore, how do the results of this modelling work compare to the modelling by Savéan et al. 2015, which is also covering the entire Dudh Koshi river catchment?*

The simulation results from Savéan et al., 2015, Nepal et al., 2014 and Eeckman et al.,2017 can not be directly compared with these obtained for the Tauche and Kharikola catchments because they do not use the same input data. Moreover, the uncertainties due to the model structure can not be directly compared in these works. Most of the paramater set calibrated for the Dudh Koshi catchment by Nepal et al., 2014 has been used for the two catchments. However only 6 of the 30 parameters have been modified in order to improve discharge and snow cover simulations separately for the two catchements. In addition, the calibration of the three global parameters for the ISBA routing module for the Kharikhola and Tauche catchments has been compared to the values calibrated for the Dudh Koshi catchment by Savean et al., 2015. It is interesting to note that the transfer velocities calibrated for the Dudh Koshi catchment are shorter than for the Tauche and the Kharikhola catchments. However this point has not been presented in this paper, in order to not dilute the main purpose.

**Specific comments:**
- *Acronym ISBA has never been explained (Interaction Sol-Biosphère-Atmosphère).*

This definition will be added on P3-L12.

- *Figure 1: maps have no geographic coordinates. Looking at figure 1D, do the authors expect that soil water storage can be an important reservoir in such a landscape?*
Coordinates will be added in fig.1. Soil depth and texture have be measured whithin the Tauche catchment and are indeed thin (max. 30 cm) and mainly sandy. In this context, soil storages are expected to be low. However, deep soil in the Tauche catchment is mainly made of ancien glaciar moraines, with large macropores. By consequence, important storage can be expected in these large macropore matrix. The behavior of such complex environment can not be directly deduced from a single picture.

- *Table 2: ET, why did the authors decide to use the method by Hargreaves and Samani 1982 and not the empirical elevation method developed for Nepal by Lambert and Chitrakar 1989 (Mountain Research and Development)?*

Hargreaves Samani method is widely used in the Himalayan region (Lutz et al. 2014) because of low requirements of input data ie. Temperature only. On the other hand, the Penmann monteith approach requires a wide range of datasets : temperature, relative humidity, sunshine hour, windspeed, The JAMS/J2000 modelling framework has some libraries for ET calculations (for example, Nepal et al 2014 used Penmann Monteith). Because of the limited data avaialble, we chose Hargreaves Samani (1982) method. Lambert and Chitrakar 1989 used available weather stations between 100-3000 meters using empirical approach

which is developed for periods 35 years ago. We wanted to benefit the approach by calculating ET by ourselves. In addition, we have also assumed that the stations data collected during our project would benefit our understanding of high-mountain hydrological processes.

- *Page 10: The discharge data is not well explained. How has the data been obtained? What are the sensors? How has stage height been converted into discharge?*

The discharge data is extensively described in Eeckman et al., 2017 (in HESS). As required by the editor's first review, material from previous work has not been repeated here. If needed, the following text could be inserted in the the Discharge data paragraph : The two hydrometric stations are equipped with Cambell$^{®}$ sensors, that record water level every 5 minutes and averaged with a 30 min.- time step. Water levels are converted into discharge through a rating curve containing 25 direct discharge measurements in Kharikhola, respectively 19 in Tauche.

- *Page 10: As for discharge, where is the data coming from? How was it recorded?*
This information is also provided in Eeckman et al., 2017 and has not been repeated. If needed, the following text could be inserted in the the Climatic data paragraph : An observation network of 10 stations (Table 1 and Fig. 1) records hourly precipitation (P ) and air temperature (T ) since 2010 and 2014. The stations are equipped with classical rain gauges and HOBO$^{®}$ sensors for

temperature. The stations are located to depict the altitudinal profile of P and T over (1) the main river valley (Dudh Kosi Valley), oriented South–North, and (2) the Kharikhola tributary river, oriented East–West.

- *Page 10/line 14: I am surprised by the temporal definition of the monsoon. Usually April to beginning of June is termed pre-monsoon and is characterized by snow melting at high elevations. The DHM normally expects the start of monsoon around the 10-15th of June*

The period between April 1 and October 30 will be renamed "summer season" and the period from November 1 to March 31 will be renamed "winter season". This definition of the summer season includes the pre-monsoon period.

- *Page 10: The precipitation interpolation method IDW needs to be better explained, to be understandable without reading secondary literature, at least the specific techniques applied in this work.*

The method developped to generate climatic input is extensively described in Eeckman et al., 2017. Upon the request of the editor's first review, this method has not been repeated in the text. If needed, the equation for precipitation interpolation (presenting the $\beta$ factor) can be introduced p10-L27.
And the following text can be inserted P10-L29 : The shape of the $\beta$ factor is controled by 5 parameters for each season ; 2 altitudinal thresholds and 3 linear lapse rates for each season. The Regional Sensitivity Analysis (Spear and

Hornberger, 1980) is applied to select the shape parameters for the $\beta$ factor that lead to acceptable precipitation fields regarding to annual simulated water budgets. Pareto fronts method is used to optimize the shape parameters in order to provide optimal bias on annual discharge for both the Kharikhola and Tauche catchments.

- *Page 11: Please give more details on the method to spatialize radiation, pressure, humidity. . . For the non-Nepal Himalayas expert audience, a more detailed explanation of the Pyramid station is appropriate.*

The following text will be inserted in P11-L4 : The hourly temperature is used to interpolate the atmospheric pressure based on the ideal gases law. The specific air humidity is deduced from the relative air humidity by combining the Wexler law and the definition of the saturating vapor pressure. The long wave radiation emitted is computed based on the air temperature using the Stefan's law.

A description of the Pyramid station will be inserted p10 L5 : the Pyramid Observatory located at 5035 m.a.s.l., whithin the Sagarmatha National Park, Khumbu region, Nepal, and managed by the association Ev-K2-CNR, Bergamo-Italy. Hourly measurements for these variables are available at the Pyramid station from October 2002 to December 2004 ( at www.evk2.isac.cnr.it/ ).

- *Page 14: Can groundwater explain the mismatch during low flow?*

Groundwater flows can indeed be an hypothesis to explain that that observed
discharge is underestimated by both models at the outlet of the Tauche catchment around November 2015 and April 2016. However, the underestimation of discharge for the Tauche catchment in 2015-2016 can also be explained by an underestimation of precipitation. Indeed, the interannual variability is hardly represented in the interpolated data set (as mentionned p11-L9), in particular because the effect of the altitude on precipitation is optimized simultenously for both years.

- *Page 15: Please discuss the soil water storage differences between the two models in more details.*

The conceptualisation of the processes in the soil are very different in both models. In ISBA, the transport equation are solved for each soil layer. The maximum soil water content is the field capacity, determined using pedotransfer functions. In ISBA, the soil water content is considered as the total volume of water contained in the soil column. In J2000, the soil depth and texture is used to determine the maximal volumes of the two soil reservoirs MacroPoreStorage (MPS) and LargePoreStorage (LPS). These soil reservoirs feed the actual evapotranspiration flux and the runoff flux. In J2000, the soil water content is considered as the sum of water stored in MPS and the water stored in LPS. These definitions of soil water storages in J2000 and ISBA are then conceptually different. In the simulation results for both models, the simulated soil water contents have different relative behavior depending on the catchment. Indeed, for the Kharikhola catchment, the soil water content simulated in ISBA gets lower than in J2000 during low flow season. On the contrary, for the Tauche catchment, the soil water content remains permanently higher in J2000 than in ISBA.

- *Page 16: It would be helpful if Dunne and Horton runoff are defined somewhere in the manuscript. Especially the differences between the two and how the two models treat these two components.*

The following text will be inserted p5-L14 :

The Dunne's flow (Dunne, 1983) and Horton's flow (Horton, 1933) are separately modelized in ISBA. The Dunne's flow is the saturation excess runoff i.e. the fraction of the precipitation that flows at the surface when the soil is saturated. The Horton's flow is the infiltration excess runoff i.e. the fraction of precipitation that flows at the surface when the intensity of the precipitation if greater than the soil capacity of infiltration. In ISBA, the Horton and Dunne flows mechanisms are modeled using the sub-grid parameterization described in Decharme and Douville (2006) : The Dunne runoff for each grid cell depends on the fraction of the cell that is saturated. The fraction of the cell that is saturated depends on the total soil water content within the cell.

The following text will be inserted P16L5 :
In ISBA, the surface overland flow is considered as the sum of the simulated Dunne's flow and Horton's flow. In J2000, the Dunne and Horton flows mechanisms are not separated and the simulated surface runoff comprises both saturation and infiltration excess runoff.

- *Page 16: The authors find that most of the water drains through the soils*

*rather than along the surface. Is this something other studies have already documented? What are the consequences of a preferential drainage through the soil compartment?*

Götz et al., 2015 (Journal of Water Resource and Protection) describe significant flow in the deep soils in the Khumbu region, in particular in unconsolidated material (e.g. ancien moraines). However, to our knowledge, no modelling work describes such important flows in the sandy soils modelled here. This result is interesting because flows in such thin and sandy soils, with such important slope are expected to be neglecticle. Moreover, this hypothesis is often made in mountains area. However, in this region, the role of flows and storages in the soil appear to be significant for surface water availability during low flow periods. This result can be particularly used by futher works focusing on land slide and/or soil and forest management.

- *Page 16 line 15: Please explain what you mean by "contribution of drainage to discharge".*

This sentence will be replaced by : The annual volume of drainage (i.e. sub-surface flow) represents 77% of the discharge at the outlet for ISBA (drainage flow at the bottom of the soil column) and 87% for J2000 (sum of the RD2, RG1 and RG2 flows).

- *Page 17: How do the snow water equivalent and snow cover findings compare to studies from the Himalayas, e.g. Wulf et al. 2016 (Advances in Water Resources)*

*or Putkonen 2004 (Arctic, Antarctic, and Alpine Research) or the two publications of the Dudh Koshi hydrology Nepal et al. and Savéan et al.?*

In Wulf et al. 2016, for the Sutlej's River basin (55000 $km^2$, from 400 m a.s.l. to 7200 m a.s.l.,3.2% glaciarized), the snow cover area is shown to be maximal in March, what corresponds to our results for the Tauche catchment (see Fig.6). In addition, for this catchment, they found a snow melt contribution to annual discharge about 35%, what corresponds to our results for the Tauche catchment. Putkonen 2004 concluded that 'only snow precipitated above 5883m'. In addition, total precipitation (rain+snow) is shown to be maximal about 3000 m.a.s.l and then decreasing. Snow fall is shown to be neglectible below 3000 m a.s.l. These results are consitant with the fact that snow fall are neglectible for the Kharikhola catchment.

Besides, in both Savean 2014 and Nepal et al. 2014, the highest snow cover in given in March-April. For the entire Dudh catchment, Savean 2014 estimated the snow melt contribution to 9% of discharge, what is lower that for the Tauche catchment. However, the Dudh Khosi catchment is about 14% glaciarized. Consequently, the results can not directly be compared to our results for non glaciarized catchments.

- *Page 18: The authors claim that for the first time they apply the models at such a high spatial and temporal resolution in mountains. Can the authors explain what is the gain of such highly-resolved modelling, especially considering that the input data is much coarser or even from different locations? Has it been tested if such a high resolution is needed or would a rather coarse resolution provide similar results. In that light, what are the errors that are propagated from interpolation of the input data through the model into the results?*

One of the main conclusion of Savean et al., 2015 is that modelling works at the regional scale in the Khumbu region presents strong uncertainties due to the heterogeneities of the region and the lack of descriptive data. Moreover, as showed in the introduction, from P2-L25 to P3-L4, large discrepancies remain in the representation of hydrological processes among several studies at a regional scale stemming from the variation in modelling applications, input data and the processes taken into account. The only seven studies quoted here provide contradictory results to explain annual water budget in the Langtang River basin and in the Dudh Koshi River basin. Besides, the responses to the water management issues in the SoluKhumbu region require a description of the hydrological systems at the local scale, tkinting into account the reality that water and hydro-energy uses only rely on small slope streams nand not on the large valley rivers. However almost all of the modeling works are set up a regional scale.

The resolution of the precipitation and temperature grids used here is 400 m, it is not coarse. In addition, the time series for other variables are the observation at the Pyramid station, which is close from both catchments. In addition, these observations have been corrected depending on the altitude.

- *Page18: If the time resolution has such an impact on the precipitation phase partition but the aim was to run both models with the same input parameters, why has the same temporal resolution not been used for both models?*

The J2000 model is not designed to run at an hourly time step. In particular, the snow melt processes are only computed at a daily time step.. In parallel, the

energy balanced in ISBA is performed at the sub-hourly time scale.

- *Page 19: In the final lines the authors state that the models can be used to predict water availability for power generation as well as under changing climate conditions. Can this conclusion really be drawn from the two very small scale studies? How can the models be scaled to different regions? What is the minimum input information needed to obtain quality results?*
This conclusion was aimed to link to modelling work to operational issues. It can be replaced by a more method-oriented conclusion : Based on this research, both models present an acceptable reliability in middle- as well as in high-mountain environments. However, the description of the surface properties is limited to the two studied catchments. The question of transferability of results to other areas then arises. The transferability of the J2000 approach has already been tested by Nepal et al., 2017 with good performances in meso-scale catchment ( 4000 km2). In addition, the description of physical properties proposed by Eeckman et al., 2017 can be extented to the majority of the surfaces in the Khumbu region. However, the parameterization of glaciarized areas should be integrated for a complete description of the environment in the Everest region.

- *Figure 5: Why is there so sharp steps in the measured discharge curve during low flow? And secondly, did it not snow in winter 2015/2016?*

As mentionned P10-L5 :
The flow at the Tauche station was frozen from 2015-01-22 to 2015-02-28 and from 2016-01-08 to 2016-02-23. Discharge is considered as nil during frozen periods. This results in sharps steps at the beginning and the end of the frozen periods. 2015-2016 was drier than 2014-2015. Neither the MOD10 data, nor the

quantitative observations made on the field present significant amount of snow during winter 2015-2016.